# Informal Institutions and Herders' Grazing Intensity Reduction Behavior: Evidence from Pastoral Areas in China

**Lijia Wang \*, Zeng Tang, Qisheng Feng and Xin Wang**

China Grass Industry Development Strategy Research Center, State Key Laboratory of Grassland Agro-Ecosystems, College of Pastoral Agriculture Science and Technology, Lanzhou University, Lanzhou 730020, China
* Correspondence: wanglijia@lzu.edu.cn

**Abstract:** Overgrazing is the key factor that has exacerbated grassland degradation in China's pastoral regions. Herder's grazing-based livestock production behavior becomes important to grassland conservation. Several formal environmental institutions and policies exist to improve grassland degradation; however, there remain contradicting conclusions regarding the contribution of these policies. Informal institutions become major instruments that might encourage herder's behavior on overgrazing. Using village rules and conventions (VRC) as a proxy for informal institutions, the article attempts to scrutinize whether the VRC emerge to respond to herders' willingness to reduce grazing intensity for grassland conservation and elicit factors affecting their reduction behavior using a Double-Hurdle model. Based on a survey of 193 respondents in Inner Mongolia and Xinjiang Autonomous regions of China, the empirical results provide evidence that VRC is effective in reducing herders' grazing intensity. In detail, the VRC in written form and an unchanging context within five years could significantly improve herders' willingness to reduce grazing intensity. Herders who consider the VRC as an important impact to their livestock production observe an increased reduction degree of grazing intensity. Additionally, variables referring to herder's education and religious belief play a significant role in the reduction degree of grazing intensity. Our findings highlight the importance of VRC in controlling herders' overgrazing behavior.

**Keywords:** informal institutions; village rules and conventions (VRC); reduction degree of grazing intensity; herder; China

## 1. Introduction

China has around 393 million hectares of grasslands, accounting for 41.7% of China's land area and 12% of the world's grasslands [1–3]. Grassland is an important part of the ecosystem and provides the basis for livestock production and herders' livelihood [4]. Approximately 17 million herders live on grasslands livestock grazing [5]. However, in recent decades, the grassland ecological environment in China has continued to deteriorate. It is estimated that around 90% of China's grasslands were degraded to some extent by the 2000s [6], and about 2 million hectares of grassland deteriorate annually [7].

A large amount of studies in the literature underline that the reasons for grassland ecosystem degradation are attributed to climate change and intensive human activities, especially in arid and semiarid areas [8–10]. Particularly, overgrazing is widely considered to be the domain factor that has exacerbated grassland degradation [11–15]. Giving the trend of grassland ecology deterioration in China's pastoral areas, a series of environmental protection policies and programs have been introduced, such as the Returning Grazing Land to Grassland Program launched in 2003 aiming at facilitating restoration of grassland vegetation by sowing grass on severely degraded grasslands, and the Grassland Ecological Compensation Policy (GECP) started in 2011 with the goals of restoring grassland ecosystems and raising herder's family income [16–18]. The main goal of these policy

interventions is grassland conservation by reducing the livestock population of China's pastoral regions.

The implementation of the GECP brings opportunities and challenges to herders' livestock breeding. The literature on the implementation effect of GECP can be divided into three aspects including ecological, economic and social performance. Researchers agree that the policy has a positive ecological effect, such as grassland ecology being generally improved [19], and the coverage and biomass of grassland vegetation being increased [20]. The obvious benefits on economic and social effects, such as raising herders' family income, improving their efficiency of husbandry, and reducing poverty, were achieved [21–23]. However, reviewing the literature to date, important questions about herder's overgrazing are still unsolved. In particular, two strands of literature exist that contribute to the understanding of the GECP effect on overgrazing. Several survey results summarized that the livestock number in pastoral area decreased [24], while other scholars generally agreed that grazing restriction policies were ineffective in controlling overgrazing, as pastoralists continue to graze illegally [25–27]. Qiu et al. (2020) claimed that higher compensation levels in the grazing ban areas caused increased herder's grazing intensities [3]. These contradicting conclusions regarding the contribution of the policy on overgrazing could to some extent be explained by the government failure, which promoted us to explore other policy tools that might motivate herders' behavior on grassland conservation. Earlier reports suggest the need to modify the top–down pattern of implementation of grassland conservation policies [28].

From the perspective of institution, the institutional environment affects human behavior [29], particularly because there are formal and informal factors motivate people to lead and carry out different activities [30]. Herders are the main operators of grassland, and their behavior directly affects grassland condition. The above-mentioned literature tells us that well-defined rules of law and political constraints matter for grassland conservation. However, Boettke et al. (2008) point out that formal rules can only be successful in promoting economic development if they are mapped onto the informal institutions [31]. In fact, institutional arrangements are the combination of formal constraints, informal rules, and their enforcement characteristics [29]. Formal rules comprise constitutional constraints, statutory rules, as well as other political constraints [32]. Informal rules, which are not designed or enforced by government, are derived from the private constraints of norms, culture, and customs which emerge spontaneously [33]. They pass through various transport mechanisms, such as imitation, oral tradition and teaching, from one generation to another [34].

Previous studies in the literature on the importance of informal institutions on grassland management have been mainly concentrated on the impact of grassland degradation. Schermer et al. (2016) stated that the informal institutions, such as normative elements and cultural–cognitive elements, result in common beliefs and share logics of actions; cultural values particularly influenced farmers' practice on ecological environment [35]. Kasymov and Thiel (2019) addressed that herders began referring to informal rules to pursue their interests [36]. Cultural factors were found to be associated with a reduced likelihood of grassland degradation [37]. The typical cultural force affecting land use might be the degradation of traditional cultural values that conserved and protected grasslands [38]. The bottom–up feedback and village-level governance should be strengthened as a policy inspection [39]. In all known self-organized resource governance mechanisms, participants delicate resources to monitor and sanction each other's behavior to reduce the possibility of free riding [40]. Ample evidence indicates that local communities can use natural resources together on a sustainable basis [41,42]. Villagers have a better understanding of their needs and concerns, especially the grassland environment on which they depend. As local institutions are the best placed to solve local problems [43], informal institutions are formed spontaneously within local communities [44]. A representative form of informal institutions is village rules and conventions (VRC), which are known as "Cun Gui Min Yue" in local language. VRC, which can be seen as self-governance rules at the grassroots

level for "self-management, self-education and self-service", has been tasked to make by-laws or to amend by-laws to suit the village demands and situation. It not only provides more elaborate plans for implementing national laws and policies but also standardizes villagers' behaviors and provides settlement rules for village affairs [45]. Village residents have room to engage in giving opinions and views during the process of making and changing those by laws whenever necessary [45]. In addition, the findings reported in Han (2018) indicated that the "tragedy of the commons" caused by the villagers' failure to abide by the VRC and the loss of collective action ability is the key to the deterioration of the grassland [10]. Therefore, VRC can be seen as an important indicator to measure herders' grassland conservation behavior, and we propose that VRC might lead to herders reducing their grazing intensity.

Extensive studies carried out a recent decade stressed the positive impact of VRC on environmental and ecological protection in rural China [46,47]. A recent study by Li et al. (2021) has found the effectiveness of informal governance on grassland quality improvement, which was based on a survey of 358 households in 60 villages in the pastoral regions of Qinghai and Gansu provinces of China [44]. Yet, few studies to date have explored how herder's grazing intensity reduction behavior responses to informal institution, i.e., village rules and conventions. Hence, in our opinion, there is a need for adequate and sufficient micro-analysis to better understand how herders' overgrazing behaviors react to informal institutions. Compared with the existing studies, this paper makes three contributions to the literature. First, informal institutions are always beyond government regulations and not part of a written legal framework [48]. In this research, different types of village rules and conventions, including oral and written, make it possible for us to investigate the effectiveness of VRC in different forms on herders' overgrazing behavior. It contributes to the literature on informal institutions in pastoral regions. Second, our study collected household data and herders' attitude toward informal institutions which might help make up the short of formal eco-environmental policies. Third, our findings would help local government to adjust grassland conservation strategies according to herder demand.

The paper attempts to investigate whether informal institutions affect herders' overgrazing behavior and to what extent they influence this behavior in pastoral China. Note that due to the space restrictions, the research does not refer to the wider religious and cultural context in which the described informal institutions were embedded and which might similarly affect herders' grazing intensity reduction behavior. Instead, we particularly focus on the analysis on the implications of village rule and conventions, which is an essential aspect of informal institution on herders' overgrazing behavior. With this narrow focus, we scrutinize the implication of the existence of VRC on herders' grazing intensity reduction behavior.

## 2. Data and Method

### 2.1. Data

The data for the research were collected during June–August 2018 in a face-to-face questionnaire in Xinjiang Uygur Autonomous Region (hereafter, Xinjiang) and Inner Mongolia Autonomous Region (hereafter, Inner Mongolia), China. The grassland area in Inner Mongolia with the largest grassland area in China is 378,300 hectares [49]. It accounts for 31.9% of the total grassland in China [41]. The grassland area in Xinjiang is 172,500 hectares [49]. The two regions are typical and traditional pastoral provinces in China where grazing is the domain agricultural activity and source of income for people living in these areas.

Stratified sampling and purposive sampling methods were adopted to conduct the survey. Firstly, a stratified sampling method is used to select sample administrative regions, sample Counties/Qi and sample townships/towns. Each autonomous region is divided into large, medium and small tiers according to the scale of animal husbandry production, and each tier randomly selects 1 to 2 cities/prefectures. Specifically, 12 cities/prefectures are under the jurisdiction in Inner Mongolia. The Year-end number of large livestock

obtained from Inner Mongolia Autonomous Regional Bureau of Statistics is applied as a proxy of the scale of animal husbandry production. Stata 16 for Windows(64-bit) (Stata-Corp LLC 4905 Lakeway Drive College Station, TX 77845 USA) is used to conduct statistical analysis on the Year-end number of large livestock. It is found that the cumulative frequency and cumulative percentage of the year-end number of large livestock below 500,000 units and above 1,000,000 units are at the two peak points of the cumulative distribution map of all cities/prefectures. One city/prefecture is selected in each scale tier. Likewise, 4 cities/prefectures are selected from Xinjiang. Next, 1 to 2 counties/Qi are randomly selected from each of the large, medium and small tiers of cities/prefectures. Secondly, considering the aim of this research is to explore the impact of VRC on herders' grazing intensity reduction behavior, in order to control the formal institutions related to grassland conservation (such as Grassland Ecological Compensation Policy, GECP) that might have an impact on herders' grazing intensity reduction behavior, purposive sampling method is thus adopted to select sample villages/Gachas in where the herders' living is fully covered by the GECP. Finally, 193 herders from 7 cities/prefectures were obtained (Table 1).

**Table 1.** Sample numbers and distribution.

| Autonomous Region | City/Prefecture | County/Qi | Number of Herders |
|---|---|---|---|
| Xinjiang Uygur Autonomous Region | Altay, Changji Hui Autonomous Prefecture, Hami City, Ili Kazakh Autonomous Prefecture | Qinghe County, Jimunai County, Mulei Kazakh Autonomous County, Barkol Kazakh Autonomous County, Nilek County, Zhaosu County | 153 |
| Inner Mongolia Autonomous Region | Hulunbuir City, Ulan Chabu City, Xilin Gol League | New Barhu Left Qi, New Barhu Right Qi, Sunite Left Qi, Xiwuzhumuqin Qi, Siziwang Qi | 40 |
| Total | 7 | 11 | 193 |

A structured questionnaire was designed to gather a range of information covering four sections: (1) herders' socio-demographics; (2) grassland management and livestock breeding; (3) implementation of village rules and conventions in surveyed regions; (4) herders' grazing situations. Content analysis was used in analyzing qualitative data gathered basing on specific themes. Household characteristics include the age and religious belief of the household head. Household-specific education is controlled for with the formal educational years that the household head experienced. Household health and mandarin level is controlled for with a five-point Likert variables range from very bad to very good for the household head (see Table 2). The number of people in the household and the total income of the household covering livestock husbandry income and non-pastoral income are included as well.

Normalized Difference Vegetation Index (NDVI) is used to measure grassland quality in the study. NDVI is constructed based on infrared and near-infrared channel remoting sensing images and has been largely used as an indicator of vegetation coverage [50]. Since grassland ecosystems have a relatively simple ecological structure, it is a feasible method to study grassland vegetation dynamics by employing these images [18]. The original NDVI data were obtained from the MOD13A3 product from NASA Earth data for the period of 2010–2020 in combination with GPS coordinated of households to create household level NDVI. Figure 1 demonstrates that the grassland quality calculated by NDVI varies from 2010 to 2020 in sample regions. NDVI was 0.377 in 2020, which is 12.1% higher than that in 2010.

**Table 2.** Socio-demographics distribution of herders in the sample.

|  |  | Sample Size (Persons) | Percent (%) |
|---|---|---|---|
| Gender | Male | 100 | 51.8 |
|  | female | 93 | 48.2 |
| Age | 18–40 years | 102 | 52.8 |
|  | 41–65 years | 78 | 40.4 |
|  | >66 years | 13 | 6.7 |
|  | Illiteracy | 13 | 6.7 |
|  | Primary school | 75 | 38.9 |
| Education | Junior high | 54 | 28.0 |
|  | High sch. or equivalent | 26 | 13.5 |
|  | Undergraduate | 22 | 11.4 |
|  | Graduate or advanced | 3 | 1.6 |
| Belief | Yes | 46 | 23.8 |
|  | No | 147 | 76.2 |
|  | Very bad | 8 | 4.1 |
|  | Somehow bad | 20 | 10.4 |
| Health | Have no particular feeling | 23 | 11.9 |
|  | Somehow good | 53 | 27.5 |
|  | very good | 89 | 46.1 |
|  | Very bad | 60 | 31.1 |
|  | Somehow bad | 26 | 13.5 |
| Mandarin | Have no particular feeling | 28 | 14.5 |
|  | Somehow good | 10 | 5.2 |
|  | very good | 69 | 35.8 |
| Hincome | <20,000 RMB | 20 | 10.4 |
|  | 20,000–50,000 RMB | 62 | 32.1 |
|  | 50,001–100,000 RMB | 57 | 29.5 |
|  | 100,001–200,000 RMB | 35 | 18.1 |
|  | >200,000 RMB | 19 | 9.8 |

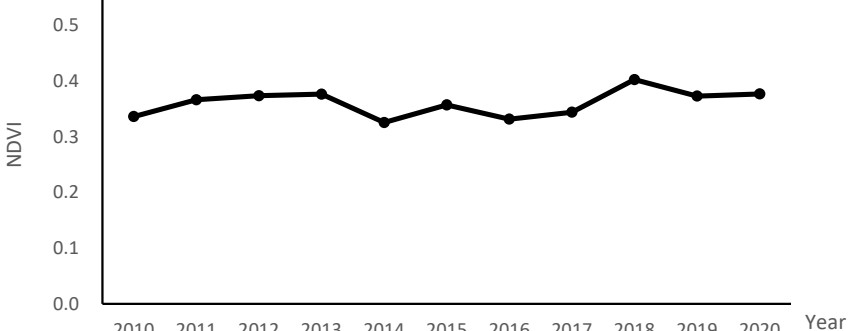

**Figure 1.** Grassland quality.

Considering that the grazing intensity significantly affects above-ground biomass [51], in our questionnaire, we recorded the number of each type of livestock on a grazing farm and grassland area of sample family from 2016 to 2018. The dependent variable "reduction degree of grazing intensity (RD)" was calculated as follows:

$$RD = (GI_{2016} - GI_{2018})/GI_{2016} \times 100\% \qquad (1)$$

where $GI_{2016}$ and $GI_{2018}$ represents the grazing intensity in 2016 and 2018, respectively. The grazing intensity was calculated by dividing the sum of all animals in sheep equivalent units by total grazing grassland area.

*2.2. Methods*

The purpose of the study is to figure out in what degree herder's grazing intensity reduction behavior is affected by VRC. The double-hurdle model which is introduced by

Cragg (1971) [52] was used in the study. Herder's reduction degree of grazing intensity is applied as a proxy for grassland conservation. The model postulates that herders must pass two separate hurdles before they are observed with a positive decrease degree of grazing intensity [53]. First, a herder becomes a "potential participant" after crossing the "first hurdle". Given positive choice, socio-economic and grassland ecological scenarios would lead to actual behavior, which is termed the "second hurdle".

Let $H_i$ be the ith herder's reduction degree of grazing intensity; then, the probability of herder choosing not to reduce grazing intensity ($H_i = 0$) is expressed as [52]:

$$\text{Prob}(H_i = 0) = \Phi(-\gamma_1' X_i') \tag{2}$$

where $\Phi$ denotes the standard normal density function; $X_i$ expresses a vector for herder ith socio-economic characteristics, grassland ecological scenarios, and VRC situations; $\gamma_1$ represents a vector of coefficients.

The second hurdle evaluates the effect of independent variables containing VRC situation variables and control variables on $H_i$ given that $H_i > 0$. It is with respect to the reduction degree given that the herders' willingness to reduce grazing intensity has decided to reduce. The distribution of $H_i$ conditional on being positive is truncated at zero with mean $\gamma_2 X_i$ and variance $\sigma^2$. The second-hurdle function can be specified as follows:

$$L(H_i \mid H_i > 0) = (1/\sigma)\Phi[(H_i - \gamma_2' X_i)/\sigma)]/\Phi(-\gamma_2' X_i/\sigma) \tag{3}$$

where $\gamma_2$ represents a vector of coefficients. A likelihood ratio statistic is employed to test the hypothesis that herders' willingness to reduce grazing intensity and their reduction behavior is independent decision.

In particular, in the situation where we have lower censoring at zero, two patterns of behavior including zero observations and positive observations exist. The sample log-likelihood formula is integrated by combining contributions for each pattern as follows:

$$logL = \sum\nolimits^n_{i=1} [I_{Hi=0}\ln\{\Phi(-(X_i'\beta)/\sigma)\} + I_{Hi>0} \ln\{(1/\sigma) \phi((H_i - X_i'\beta)/\sigma)\} \tag{4}$$

where $I$ denotes the indicator formula; if the subscripted expression is true, $I$ takes the value one. $\Phi$ represents the standard normal cumulative, and $\phi$ represents the probability density function. $logL$ is maximized in terms of the parameters included in the vector $\beta$ and the standard deviation parameter $\sigma$.

Notably, the problem with herder's reduction degree of grazing intensity is that its distribution is irregular. If used directly as a response variable, it may cause inconsistency and non-normality of error terms [54]. In this research, we used the logarithm of positive reduction degree of grazing intensity since the transformed variable is more prone to be normally distributed. Figure 2 depicts the histograms of both original and transformed herder's reduction degree of grazing intensity. Additionally, the logarithm transformation of the response variable is more amenable in computing elasticity of reduction degree of grazing intensity with respect to demographic variables.

### 2.3. Descriptive Statistical Analysis

Table 3 presents the definition and description of the variables used in the study along with the mean differences for herders with willingness to reduce grazing intensity and those unwilling to reduce grazing intensity. Around 76.2% of interviewed herders were willing to reduce grazing intensity to protect grassland ecology. These are labeled as "perceivers" in Table 3. Note that the variable "duration of VRC" is directly obtained from the questionnaire. We asked the respondent to answer the question, "In which year the VRC was established in your village no matter in oral or written form?" With reference to herders' social–psychological characteristics, Table 3 expresses that the mean levels of herders' mandarin and household size were significantly higher among the perceivers than non-perceivers, while religious belief was more pronounced among non-perceivers.

Moreover, male herders showed a significantly greater unwillingness to reduce grazing intensity than female herders.

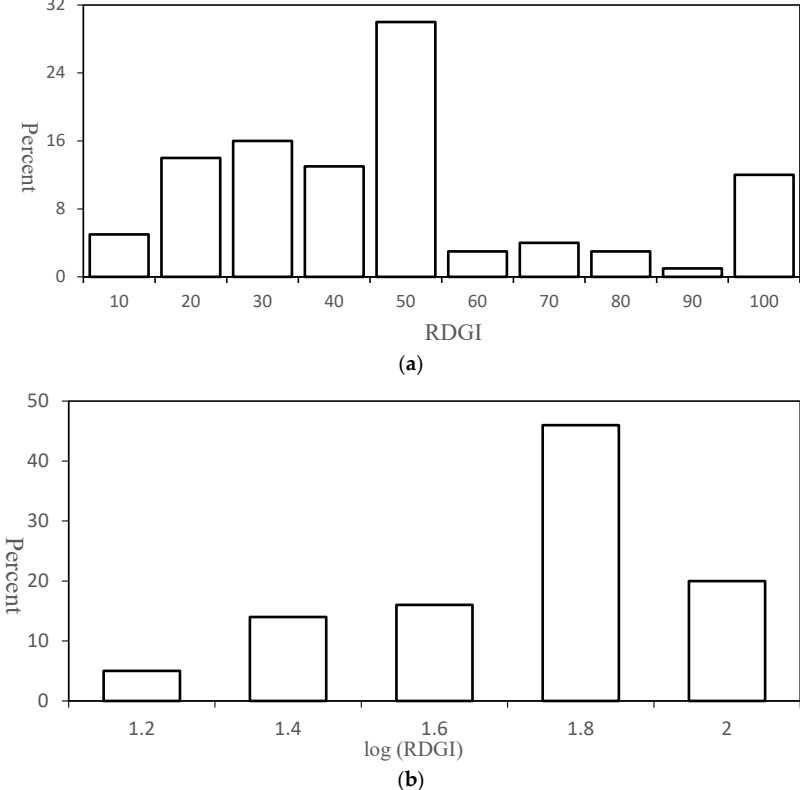

(**a**)

(**b**)

**Figure 2.** Distribution of reduction degree of grazing intensity at the original scale and logarithm transformed scale (for positive percentage). (**a**) Reduction degree in original scale (in percent); (**b**) Reduction degree in logarithm transformed scale.

VRC has different types (Table 4). Only 64.2% respondents told us that the VRC values in their village were in written forms, while more than one-third were in oral forms. Over half of the herders (51.3%) reported that the content of the VRC had been changed with the recent five years.

Figure 3 depicts the distribution of herder's reduction degree of grazing intensity. Nearly half of the herders (45.6%) mentioned that their reduction degree of grazing intensity was between 31 and 60%. Around one-third of the observations reported a reduction degree of grazing intensity from 10 to 30%. In all, 12.2% of the respondents were identified that their reduction degree of grazing intensity reached over 91%.

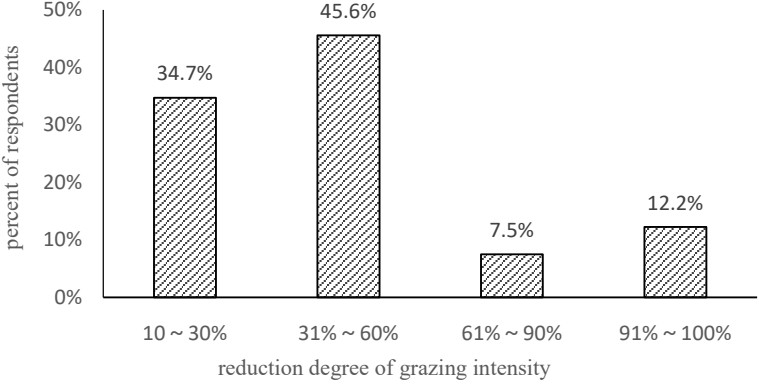

**Figure 3.** Distribution of herder's reduction degree of grazing intensity.

**Table 3.** Variable definitions and summary statistics.

| Variables | Description | Variable Scale | Mean | | *p*-Value |
|---|---|---|---|---|---|
| | | | Perceivers (n = 147) | Non-Perceivers (n = 46) | |
| WTR | Willingness to reduce grazing intensity | 1 = yes, 0 = otherwise | 1 | 0 | - |
| RD | Reduction degree of grazing intensity | % | 47.755 | - | - |
| Time | Duration of VRC | Years | 6.748 | 6.565 | 0.868 |
| Written | VRC in written communication | 1 = yes, 0 = otherwise | 0.673 | 0.543 | 0.110 |
| Change | VRC had changed within 5 years | 1 = yes, 0 = otherwise | 0.463 | 0.674 | 0.012 ** |
| Impact | VRC had important impact on herder's husbandry production | 1 = yes, 0 = otherwise | 0.8027 | 0.6522 | 0.035 ** |
| | | Control variables | | | |
| Gender | Gender | 1 = male, 0 = female | 0.483 | 0.630 | 0.082 * |
| Age | Age of the household head | Years | 40.830 | 41.848 | 0.699 |
| Education | Education level | Years | 7.476 | 7.304 | 0.815 |
| Belief | Religious belief of household head | 1 = yes, 0 = otherwise | 0.184 | 0.413 | 0.001 *** |
| Health | Health level of household head | 1 = very bad, 2 = somehow bad,3 = have no particular feeling, 4 = somehow good, 5 = very good | 4.041 | 3.913 | 0.520 |
| Mandarin | Mandarin level of household head | 1 = very bad, 2 = somehow bad,3 = have no particular feeling, 4 = somehow good, 5 = very good | 3.136 | 2.609 | 0.066 * |
| Hsize | Household size (the number of people in a household) | person | 4.517 | 4.130 | 0.083 * |
| Hincome | Household income | Ten thousand yuan | 9.078 | 9.317 | 0.880 |
| Landtype | Landform type | 1 = flat; 2 = slope | 1.891 | 1.717 | 0.022 ** |
| NDVI | Normalized Difference Vegetation Index | - | 0.363 | 0.362 | 0.986 |
| Graze | Grazing intensity | Sheep equivalent units/ha. | 2.025 | 2.355 | 0.591 |

Note: number of observations n = 193. Yuan is Chinese currency. Exchange rate: 1 US$ = 6.61 RMB in 2018
* *p* < 0.10, ** *p* < 0.05, *** *p* < 0.01.

**Table 4.** Number of herders with VRC in 2018 by different types.

| VRC Types | Sample Number | Percent in Sample |
|---|---|---|
| All with VRC on grazing intensity and production | 193 | 100 |
| Types of communication | | |
| 　Written | 124 | 64.2 |
| 　Oral | 69 | 35.8 |
| Having changed within 5 years | | |
| 　Yes | 99 | 51.3 |
| 　No | 94 | 48.7 |

　　　Table 5 presents the summary statistics for attitude variables that were partially used in the double-hurdle model. Three questions which are measured by five-point Likert scale were asked to examine herders' attitudes toward VRC. Specifically, in terms of the variable

"Impact of VRC on promoting harmony among herders", we requested herders to evaluate the impact degree of VRC on their daily life activities and their relationship with neighbors. With respect to the variable "Impact of VRC on herder's husbandry production", herders needed to rate in what degree their grazing behavior was affected by the context of VRC. The degree of herder's compliance with VRC was measured by the level of compliance that herders have toward the rules and conventions in VRC. Note that for the purpose of presentation, the five-point Likert scale classification was merged into three: "high" in Table 5 combines the frequency of "very high" and "high" responses; "low" in Table 5 combines the frequency of "very low" and "low" responses.

**Table 5.** Herder's attitude toward VRC (percent of respondents).

| Items | Low | Moderate | High |
|---|---|---|---|
| Impact of VRC on promoting harmony among herders | 16.2 | 11.4 | 81.3 |
| Impact of VRC on herder's husbandry production | 12.5 | 10.9 | 76.7 |
| Degree of herder's compliance with VRC | 2.1 | 4.7 | 93.3 |

According to Table 5, 81.3% of the respondents admitted that VRC could highly promote the harmony among villagers, while 76.7% of the respondents told us that their living style and production activities were highly influenced by VRC. With regard to the compliance with VRC, over 90% of herders (93.3%) claimed to conscientiously comply with VRC. According to our face-to-face interview, herders' acceptance and recognition of VRC was relatively high. When we asked them, "Did you feel unfair of the content of VRC?" Most of the herders claimed with no feeling of unfairness, as they thought that all the villagers should follow the same rules.

## 3. Results and Discussion

### 3.1. Multicollinearity Analysis

It is difficult to assess the relative importance in determining some dependent variables when two supposedly independent variables are highly correlated [55]. Thus, multicollinearity analysis is applied before the estimation of the double-hurdle model. The result shows that there is multicollinearity between variables "household income" and variable "grazing intensity". In addition, variable "grazing intensity" is as similar as dependent variable *RD*, thus, the two variables are excluded from the double-hurdle model.

### 3.2. Double-Hurdle Models

The estimation results of the impact of VRC on herders' willingness to reduce grazing intensity and their reduction degree of grazing intensity are reported in Table 6. The first two columns show the effects of the VRC characteristics and herder demographics on the probability that a herder expresses willingness to reduce grazing intensity, while the determinant of the reduction degree of grazing intensity is illustrated in the last two columns of Table 6. Marginal effects evaluated at the means of the explanatory variables are contained.

#### 3.2.1. Impact of Willingness to Reduce Grazing Intensity

As indicated in the double-hurdle estimates in Table 6, the duration of VRC is not a significant determinant in herder's decision on whether to reduce or how deep to reduce the grazing intensity. The VRC in written form significantly improve herder's willingness to reduce grazing intensity. Herders hold a higher compliance of VRC in written form than that in oral form. Non-compliance does not result in punishment but rather in shame, since norms are morally governed [35]. This is in consisting with the findings from Li et al. (2021) [44] that grassland quality improved when the grassroots governance was in a written form. A change for VRC negatively affects the willingness of herders to reduce

grazing intensity. According to our face-to-face interview, a majority of respondents told us that the frequent revision of the content of VRC would largely affect their compliance and trust in VRC. Moreover, herders who consider the VRC as an important impact to his/her husbandry production would have a higher willingness to reduce grazing intensity. Our research complements the growing informal institutional literature by indicating the strong effect of VRC on herders' behavior of reducing grazing intensity.

**Table 6.** Estimation results of the impact of VRC on *WTR* and *RD*.

| Independent Variables | First Hurdle Willingness to Reduce Grazing Intensity | | Second Hurdle Reduction Degree of Grazing Intensity | |
|---|---|---|---|---|
| | Coef. (Std. Err.) | Marginal Effect | Coef. (Std. Err.) | Elasticity [a] |
| | Village rules and conventions | | | |
| Time | −0.011 (0.017) | −0.003 | −0.001 (0.003) | 0.001 |
| Written | 0.405 * (0.240) | 0.115 | −0.002 (0.040) | −0.002 |
| Change | −0.684 ** (0.264) | −0.183 | −0.006 (0.041) | −0.004 |
| Impact | 0.478 * (0.253) | 0.143 | 0.120 *** (0.047) | 0.079 |
| | Control variables | | | |
| Gender | −0.569 ** (0.231) | −0.153 | −0.041 (0.038) | −0.027 |
| Age | 0.001 (0.009) | 0.0002 | −0.001 (0.002) | 0.000 |
| Education | −0.041 (0.038) | −0.011 | −0.013 ** (0.006) | −0.009 |
| Belief | −0.600 ** (0.250) | −0.183 | 0.085* (0.048) | 0.056 |
| Health | −0.006 (0.108) | −0.002 | 0.012 (0.016) | 0.008 |
| Mandarin | 0.243 ** (0.102) | 0.066 | 0.010 (0.015) | 0.007 |
| Hsize | 0.095 (0.092) | 0.026 | 0.001 (0.014) | 0.000 |
| Landtype | 0.543 ** (0.254) | 0.148 | 0.003 (0.049) | 0.002 |
| NDVI | −1.409 ** (0.635) | −0.383 | −0.419 *** (0.101) | −0.276 |
| Constant | −0.246 (1.042) | - | 1.707 (0.182) | |
| Observations (n) | 193 | | 147 | |
| Wald $\chi^2$ | 39.94 | | 47.01 | |
| Log pseudo-likelihood | −86.016 | | 17.701 | |
| Sigma | | | 0.013 | |

Note: [a] The elasticity is measured at the sample mean; * $p < 0.10$, ** $p < 0.05$, *** $p < 0.01$.

The result emphasizes the importance of household head's gender, religiosity, mandarin level, grass type and NDVI in explaining herder's willingness to reduce grazing intensity. Female household head with better mandarin level tends to be more willing to reduce grazing intensity for the purpose of grassland conservation. In combination with our face-to-face interview, a plausible explanation is that a male household head has more responsibility to the family income. They told us that cattle and sheep are the source of income. More cattle and sheep represent better wealth. Thus, they are unwilling to reduce grazing intensity. Meanwhile, there is a negative association between herders' religious belief and their willingness to reduce grazing intensity. There are several explanations for this negative association. Most notably, herders with religiosity believes that livestock are culturally important [56]. Animals in grassland not only represent wealth but also a symbol of status. In view of this, herders with religious belief are unwilling to reduce the number of cattle or sheep. Herders with religious belief hold that nomadism is conducive to protect the grassland ecology instead of destroying it.

Each respondent's mandarin level plays an important role in the herder's willingness to reduce grazing intensity, while the magnitude is small (6.65%). Herders with better mandarin levels are more likely to reduce grazing intensity. One possible reason is that herders with better Chinese proficiency would be easier to communicate with people outside of the pastoral areas and be more likely to have a better understanding of the significance of grassland conservation. Another possible explanation is that herders with better Chinese proficiency would have more chances to acquire non-pastoral employment to widen their family income channels, since diversification of the income source is

an important guarantee to meet subsistence requirements. The less a herder's family income source relies on livestock production, the higher the herder's willingness to reduce grazing intensity.

The result also demonstrates that herders whose pastures are located on sloping land are more willing to reduce grazing intensity. NDVI has a significantly negative impact on respondent's willingness to reduce grazing intensity, which is probably because higher NDVI commonly leads to an increase in aboveground biomass, which can provide forage for more cattle and sheep.

### 3.2.2. Impact on Reduction Degree of Grazing Intensity

Table 6 reports the parameter estimates and elasticity for the determinants affecting the reduction degree of grazing intensity. The elasticity expresses changes in the explanatory variables on the level of the degree of grazing intensity reduced by herders with willingness to reduce grazing intensity. As clarified earlier, herders who consider the VRC as an important impact to their husbandry production observes an increased reduction degree for grazing intensity of 3.07%. Variables referring to herder's religiosity play a positively significant role on the reduction degree of grazing intensity. In other words, household head with religious belief observes an increased reduction degree of grazing intensity of 3.14%. This result is consistent with the findings from Yang et al. (2022) [57] that ethnic group was an important factor to affect the response of herders' behavior. Another explanation can resort to the survey data, in which the ratio of herders with religiosity whose non-animal husbandry income is higher than those without religiosity (Figure 4). It means that the family income of herders with religious belief relies less on livestock production compared with herders without religious belief in sample regions. The regression result thus indicates a positive relation between religiosity and reduction degree of grazing intensity.

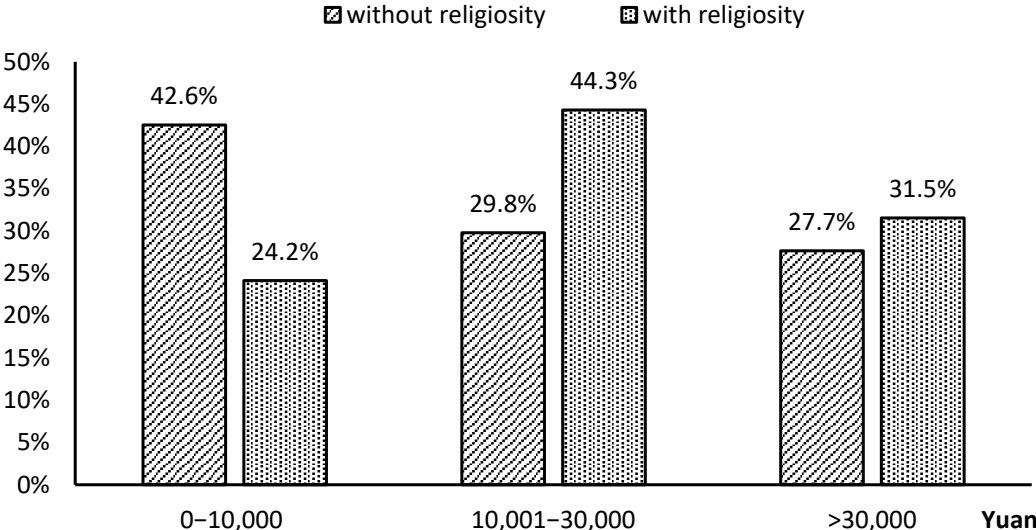

**Figure 4.** The ratio comparison between herders with religiosity and without religiosity in different levels of non-animal husbandry income.

Household heads with more schooling years observe a decreased reduction degree of grazing intensity of 0.36%. The result can be verified by our survey data that the grazing intensity started to decrease when the household head had a college degree (Figure 5). This finding is also consistent with those of Gao (2016) [58], who discovered that each additional year of education reduced herder's grazing intensity by 3.6%. Jimoh et al. (2020) suggested that the education of herders on preventing grassland degradation from overgrazing should be emphasized [39].

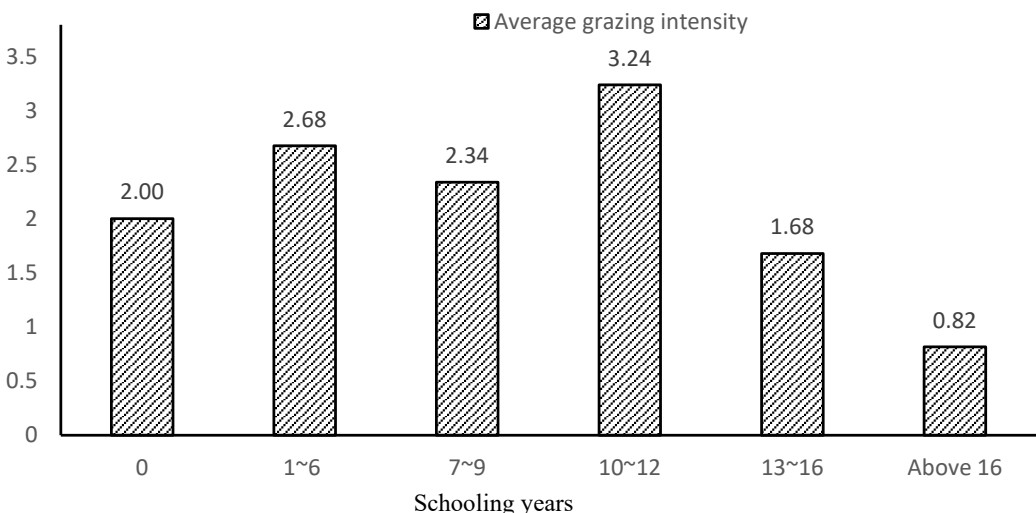

**Figure 5.** The relationship between schooling years of household head and family's average grazing intensity.

More importantly, another grazing characteristic that has a negative impact on the degree of grazing intensity reduced by herders who have shown a willingness to reduce grazing intensity is NDVI. That is, herders whose grassland is with greater NDVI observe a decreased reduction degree of grazing intensity of 6.92%. Our result corroborates the earlier literature [59] that the relative grazing intensity increased gradually with the decrease in NDVI.

## 4. Conclusions and Implications

Taking into consideration the rising position of VRC on herders' husbandry activity, the paper specifically concentrated on herders' willingness to reduce grazing intensity and investigated their reduction degree of grazing intensity, which are important for the grassland conservation and sustainable development of China's pastoral regions. Field survey data from seven cities/prefectures in two major pastoral Autonomous Regions in China are collected. Our empirical results provide evidence that to some extent, VRC is effective in reducing herders' grazing intensity. Apart from that, findings of the study can be used by policymakers to encourage herders' behavior of reducing grazing intensity, as 23.8% of herder households are unwilling to reduce grazing intensity in the sample regions. The primarily results are as follows.

Firstly, the local government should pay much more attention to the standardization of formulation and the stability of the content of VRC. In other words, VRC should be in written form at the establishment procedure. The design of VRC should be detailed and specified to ensure being unchanged within recent years, as frequent changes would affect herders' trust and loyalty to the VRC.

Secondly, specific regulations, such as overgrazing seriously endangering the grassland ecology, and thus affecting herders' livelihood, should be supplemented in VRC. Penalties to those who violate the rules in VRC should be clearly defined and strictly carried out, as the penalty form of grassroot governance indicated an increase in grassland quality [44].

Thirdly, religiosity has a profound impact on herders' spirit and behavior. With the consideration of respecting herders' religiosity, religious organizations should be properly guided to combine the practice of grassland conservation to herder's livelihood, i.e., supplementing regulations related to grassland conservation and inculcating the concept of sustainable development. As indicated in previous literature, religiosity ties herders together in communities and introduces mechanisms for the enforcement of desirable behavior [60].

Furthermore, regular Chinese training for herders to improve their Chinese proficiency is highly recommended. For instance, Chinese evening schools and training classes should be established to promote herders' ability to obtain non-farm employment and thus to reduce the dependence of their family income on livestock breeding.

Finally, digital technology such as cameras, smart phones, GPS satellite positioning systems and UAV monitoring systems can be used to monitor cattle and sheep for herders with large pastures. It is also a promising tool to estimate pasture quality parameters and biomass availability. The accurate data obtained from digital technology can be useful for herders to keep the number of livestock at a reasonable level instead of overgrazing.

We have sought to offer a new research perspective on the impact of herder's grazing intensity reduction in pastoral areas. It is encouraging to find out that informal institutions, such as village rules and conventions, contribute positively to herder's grazing intensity reduction in the pastoral regions. We believe that the findings and implications can provide useful reference for grassland protection in global drylands in developing countries, such as Ethiopia, Eastern and Southern Africa, as well as Argentina. In total, 90% of dryland inhabitants live in developing countries [61]. Overgrazing by domestic livestock is a principal anthropogenic force leading to their desertification [62]. These countries provide several government aids but fail to make a significant impact on the overgrazing situation [63]. Thereafter, in these dryland countries, the government should pay much more attention to the potential effect of informal institutions on herders' grazing behavior and promote establishing VRC by herder-selves to manage their overgrazing behaviors.

Much theoretical and empirical work remains to be done. In the body of this paper, we have only focused on village rules and conventions, while informal institutions broadly include religion, norms, culture, and customs [64,65]. Further research would explore the other aspects of informal institutions and specify their impact on herders' grazing intensity reduction more explicitly. Additionally, the double-hurdle model assumes the shocks to the willingness and reduction degree processes being independent, which is not always realistic [66]. Thus, the econometric model should be further improved.

**Author Contributions:** Conceptualization, L.W.; Investigation, Z.T. and X.W.; Software, L.W. and X.W.; Data curation, L.W. and Z.T.; Writing—original draft preparation, L.W.; Writing—review and editing, L.W., Z.T. and Q.F. All authors have read and agreed to the published version of the manuscript.

**Funding:** This work was founded by the Chinese Academy of Engineering (2020-XZ-29; 2022-HZ-09) and supported by the Fundamental Research Funds for the Central Universities (lzujbky-2021-kb13).

**Institutional Review Board Statement:** Not applicable.

**Informed Consent Statement:** Not applicable.

**Data Availability Statement:** The data presented in the study are available upon request from the corresponding author.

**Acknowledgments:** The authors would like to thank the three anonymous reviewers for constructive and helpful comments.

**Conflicts of Interest:** The authors declare no conflict of interest.

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
