# Peer review of "Informal Institutions and Herders’ Grazing Intensity Reduction Behavior: Evidence from Pastoral Areas in China"

_land, doi:10.3390/land11091398_

Round 1

Reviewer 1 Report

The paper applies the Double-Hurdle model to empirically examine the impact of informal institutions on grazing intensity reduction behavior, based on survey data of 193 herders sampled from two main provinces of pastoral areas. Generally, the paper has a well-defined structure and a smooth flow. The empirical strategy that validates the measurement of informal institutions is interesting. The findings are rational and can provide insights to solve the problem of overgrazing in pastoral areas. Nevertheless, I have some minor comments.

1. Motivation. This paper should further discuss the importance and significance of paying attention to Informal Institutions and Herders’ Grazing Intensity Reduction Behavior in the Introduction.

2. The data in this paper show that 100% of villages have village rules and regulations, which is inconsistent with the conclusion of Li et al. (2021) found that 67 percent of villages had established village regulations in 2017. Why?

3. Table 2. The dependent variable, such as the Reduction degree of grazing intensity, is not well defined. Meanwhile, the measurement of Time, Written, Change, and Impact for VRC has not been explained. It is suggested to add a measurement description of some core explanatory variables.

4. Variable selection. Why choose graze intensity as a control variable? I think this variable is the same as the dependent variable and should not be used as a control variable. Also, why choose NDVI as a control variable? Please give a reasonable explanation.

5. Table 4. How to measure herder’s attitude towards VRC? A detailed explanation is needed.

6. Table5. The significance should be placed after the regression coefficient rather than after the parentheses; The meaning of the numbers in parentheses is not given. In addition, why is a change for VRC negatively affecting the willingness of herders to reduce grazing intensity? And why do beliefs negatively affect the willingness of herders to grazing intensity while positively affecting the reduction degree of grazing intensity? All these findings need more explanation.

7.P10. the title of Section3.3.1 should be changed to “impact of willingness to reduce grazing intensity”.

8. Conclusion: What are the implications of the conclusion for other developing countries? These issues need to be further elaborated on in the conclusion.

9. The English writing of the paper is not good enough. Please further edit and polish the writing.

Reviewer 2 Report

-          Clarify your question or hypothesis in the introduction

-          “large, medium and small tiers according to the scale of animal husbandry production”. This needs clarification. How have you differentiated/classified them? How subjective was this classification and to what extent has it affected the randomness?

-          “a typical sampling method is thus adopted”. Namely?

-          You carried out analytic and not descriptive statistics. Is the randomness really necessary?

-          Try to make the conclusions more concise and easy to read. Point out certain non-trivial findings

Reviewer 3 Report

Dear Authors,

The submitted manuscript titled „Informal Institutions and Herders’ Grazing Intensity Reduction Behavior: Evidence from Pastoral Regions in China” contains interesting findings. However, I have fund some flawns, which-in my opinion- should be improved before an eventual publication.

1.       Lines 108-112 I suggest to list the specific aims of investigations instead of description of particular sections of manuscript.

2.       I think that the description of study area should be added in section Material and methods. Such description should present the area and type of grasslands, as well as the characteristics of tratitional use.

3.       In my opinion the chapter Result and discussion should be divided into two separate sections. Then both sections would be be more suitable to follow.

4.       In section Results the structure of respondents (regarding to age, gender, education etc.) should be presented e.g. on charts .

5.       The chapter Conclusions soul contain the proposition of further directions of studies.

6.       In my opinion the interesting results are poorly compared and discussed with other findings contain in numerous literature sources. I entourage Authors to look into below listed papers, perhaps some of them would be useful in the manuscript improvements:

·         Zhang, J., Zhang, L., Liu, W. et al. Livestock-carrying capacity and overgrazing status of alpine grassland in the Three-River Headwaters region, China. J. Geogr. Sci. 24, 303–312 (2014). https://doi.org/10.1007/s11442-014-1089-z

·         Saheed O. Jimoh, Xiu Feng, Ping Li, Yulu Hou, Xiangyang Hou. 2020. Risk-Overgrazing Relationship Model: An Empirical Analysis of Grassland Farms in Northern China, Rangeland Ecology & Management, 73( 4),463-472, https://doi.org/10.1016/j.rama.2020.03.006

·         Jushan Liu, Forest Isbell, Quanhui Ma, Ying Chen, Fu Xing, Wei Sun, Ling Wang, Jian Li, Yunbo Wang, Fujiang Hou, Xiaoping Xin, Zhibiao Nan, Nico Eisenhauer, Deli Wang,

·         Overgrazing, not haying, decreases grassland topsoil organic carbon by decreasing plant species richness along an aridity gradient in Northern China. 2022. Agriculture, Ecosystems & Environment, 332,107935, https://doi.org/10.1016/j.agee.2022.107935.

Round 2

Reviewer 3 Report

Dear Authors,

In my opinion Your manuscript has received the satisfactory corrections. therefore I do not have any further suggestions of changes.